# Diets of the Barents Sea cod (*Gadus morhua*) from the 1930s to 2018

Bryony L. Townhill[1], Rebecca E. Holt[2], Bjarte Bogstad[3], Joël M. Durant[2], John K. Pinnegar[1,4], Andrey V. Dolgov[5,6,7], Natalia A. Yaragina[5], Edda Johannesen[3], Geir Ottersen[2,3]

[1] Centre for Environment, Fisheries and Aquaculture Science (Cefas), Pakefield Road, Lowestoft, Suffolk, NR33 0HT, UK

[2] Centre for Ecological and Evolutionary Synthesis, Department of Biosciences, University of Oslo, P.O. Box 1066 Blindern, 0316 Oslo, Norway

[3] Institute of Marine Research, P.O. Box 1870 Nordnes, 5817 Bergen, Norway

[4] Collaborative Centre for Sustainable Use of the Seas (CCSUS), University of East Anglia, Norwich NR4 TJ, UK

[5] Polar branch of the Federal State Budget Scientific Institution "Russian Federal Research Institute of Fisheries and Oceanography" (VNIRO, formerly PINRO), 6, Academician Knipovich Street, Murmansk 183038, Russia

[6] Federal State Educational Institution of Higher Education "Murmansk State Technical University", 13, Sportivnaya Street, Murmansk, 183010, Russia

[7] Tomsk State University, 36, Lenin Avenue, 634050 Tomsk, Russia

*Correspondence to: Bryony L. Townhill (bryony.townhill@cefas.co.uk)*

29

30

**Abstract**

A new dataset on the diet of Atlantic cod in the Barents Sea from the 1930s to the present day has been compiled, to produce one of the largest fish diet datasets available globally. Atlantic cod is one of the most ecologically and commercially important fish species in the North Atlantic. The stock in the Barents Sea is by far the largest, as a result of both successful management and favourable environmental conditions since the early 2000s. As a top predator, cod plays a key role in the Barents Sea ecosystem. The species has a broad diet consisting mainly of crustaceans and teleost fish, and both the amount and type of prey vary in space and time. The data, from Russia, Norway and the United Kingdom, represents quantitative stomach contents records from more than 400,000 fish, and qualitative data from 2.5 million fish. Much of the data is from joint collaborative surveys between Norway and Russia. The sampling was conducted throughout each year allowing for seasonal, annual and decadal comparisons to be made. Visual analysis shows cod diets have changed considerably from the start of the dataset in the 1930s to the present day. There was a large proportion of herring in the diets in the 1930s, whereas in more recent decades, capelin, invertebrates and other fish dominate. There are also significant interannual asynchronous fluctuations in prey, particularly capelin and euphausiids. Combining these datasets can help us understand how the environment and ecosystems are responding to climatic changes, and what influences the diet and prey switching of cod. Trends in temperature and variability indices can be tested against the occurrence of different prey items, and the effects of fishing pressure on cod and prey stocks on diet composition could be investigated. The dataset will also enable us to improve parametrisation of food web models, and to forecast how Barents Sea fisheries may respond in the future, to management and to climate change. The Russian data is available through joint projects with Polar branch of VNIRO. The UK and Norwegian data (Townhill et al., 2020) is being released with this paper at doi: 10.21335/NMDC-2139169383.













## 1 Introduction

Here we document a new extensive dataset on the stomach content of Atlantic cod (*Gadus morhua*) in the Barents Sea. We have compiled the dataset by joining historical data from the UK (1930-1964) with historical data from the Soviet Union/Russia (1947-1983) and more recent (1984-2018) data from a large existing joint Norwegian-Russian cod stomach-content database. The resulting dataset spans the period from 1930 until present day.

Comprehensive information on the predation dynamics of ecologically important species, based upon the analysis of individual stomach-contents, is vital for an understanding of how the biological components in an ecosystem are connected (Hyslop, 1980; Holt et al., 2019). Such data can provide detailed knowledge on the diet of a species in an area at a particular time. When conducted over long temporal scales and across size classes, spatially high-resolution stomach-content data can provide data that is key to understanding trophic interactions in marine ecosystems.

Unfortunately, long-term high-quality fish population diet data with good spatio-temporal coverage are rare, as the effort and resources required to collect and analyse stomach-contents at this scale is considerable. However, due mainly to the stock's commercial importance, both Russia, Norway and the UK have invested considerable resources in sampling, working up and sustaining stomach-contents data for the Barents Sea (or Northeast Arctic) cod. Diet data is valuable in understanding trophic interactions; particularly important in areas where multiple species are caught. For example, diet data allows predation mortality to be included in stock assessments (ICES, 2019), and to understand inter-specific interactions between predators (Durant et al., 2014).

To support the cod fishery in the Barents Sea, the UK carried out surveys from the 1930s, mainly collecting catch and length data, but also recording stomach contents. They collected content data from between a few hundred to 3,500 stomachs each year, ending in the 1960s. The Norwegian-Russian data originate from a joint research project on the diet and food consumption of Barents Sea fish, with cod as the main study species, initiated in the mid-1980s. This was a joint endeavour between IMR (Institute of Marine Research, Norway) and PINRO (Knipovich Polar Research Institute of Marine Fisheries and Oceanography, since 2019 names the Polar Branch of the Russian Federal Research Institute of Fisheries and Oceanography (VNIRO); Mehl, 1986; Mehl and Yaragina, 1992; Dolgov et al., 2007, 2011; Yaragina et al., 2009). An average of 8,153 stomachs were analysed each year (Holt et al., 2019). In addition, there are also numerous Russian cod diet data that was collected from the 1930s-80s, (Dolgov et al., 2007; Yaragina and Dolgov, 2011) which are described here and summarized in the Supplementary Material. These could not be made available in the main dataset published with this manuscript, but are available under joint research projects.

Atlantic cod is one of the most ecologically and commercially important fish species in the North Atlantic and the Barents Sea stock is by far the largest. As opposed to many other cod stocks and other fish world-wide, the Barents Sea cod are doing well, a result of both successful management and favourable environmental conditions since the early 2000s (Kjesbu et al., 2014, Ottersen et al., 2014, Fossheim et al., 2015). Cod plays a key role in the Barents Sea ecosystem and is the dominating top predator. While cod has a broad diet consisting mainly of crustaceans and teleost fish, the amount and kind of prey actually available varies in space and time as well as by cod size (Zatsepin and Petrova, 1939; Yaragina et al., 2009, Johannesen et al., 2012, 2015; Holt et al., 2019).

For Atlantic cod, being arguably one of the most important fish on the planet, such diet data exist in several seas: e.g. the Baltic (Neuenfeldt and Beyer, 2006); on Georges Bank (Tsou and Collie, 2001); the Gulf of Maine, US (Willis et al., 2013); Icelandic waters (Pálsson and Björnsson, 2011); and the northeast US shelf ecosystem (Link & Garrison 2002). A comparison of Atlantic cod diet and the role of cod in the various ecosystems was made by Link et al. (2009). Data on the diet of other northeast Atlantic species have been recently released, allowing analysis of herring, blue whiting, mackerel, albacore and bluefin tuna diets (Pinnegar et al., 2015). The time series of these pelagic species begin in the 1860s, and combine data from France, Norway, Iceland, Ireland and the UK. Here, we compile a similar dataset of Barents Sea cod diet data, from Norway, Russia and the UK.

**2 Data and methodology**

**2.1 UK Barents Sea surveys**

A UK fishery began in the Barents Sea in 1905, with increased exploitation from 1929. Catches of cod and therefore profits were high, particularly from the 1930s when sea temperatures in the area became warmer (Cushing, 1966) and cod stock sizes increased to historic high levels in the 1930s-1940s (Hylen, 2002). Aimed at investigating the cod fishery, and the influence of temperature, which already at the time was known to influence cod distributions, the UK carried out fisheries surveys in the Barents Sea from the 1930s to the 1960s, with a break for WWII. The surveys were conducted firstly on-board commercial fishing vessels, and later with a dedicated Arctic survey vessel, the *RV Ernest Holt* (Graham, 1953). The surveys collected data on cod abundance, length distributions, temperature, salinity and depth, and samples of cod stomachs were also taken. This was less systematic than for present-day cruises, and so the data is less statistically robust than for the data for the 1980s onwards. The frequency of prey items was recorded rather than the mass of each prey item, and no data on stomach fullness was collected. For the majority of surveys, prey occurrence is recorded for each stomach individually. For some however, pooled data is provided, for up to 198 stomachs in total. The survey methodology is described in Graham (1953) and summarised in Townhill et al. (2015). On the RV Ernest Holt, a standard otter trawl was used, with and without Vigneron-Dahl gear. Rather than using a statistically designed survey grid, the scientists wanted to find large cod groups and so vessels searched for high cod catches, operating more like a commercial fishing boat.

*DAPSTOM database summary*

Under the Centre for Environment, Fisheries and Aquaculture Science (Cefas) project Trawling Through Time (DP332) and the Norwegian-British-Russian research project CoDINA (Cod: Diet and food web dyNAmics), funded by the Research Council of Norway, the data for these surveys were digitized from paper logbooks held by Cefas and the stomach data is held in the DAPSTOM database (Cefas, 2014; Pinnegar, 2014). The DAPSTOM (Database And Portal for Fish STOMach records diet database), described in Pinnegar (2019) contains 256,354 records from 360,561 stomachs, with the first records from the 1830s. These are from 204 species and 9,445 research cruises/sampling campaigns. 28% of the records are for Atlantic cod, mainly for the seas around the UK, but also including these for the Barents Sea.

**2.2 The joint Norway-Russia research programme on trophic relationships in the Barents Sea**

Russian and Norwegian surveys include cod stomach sampling in their regular procedures, as described in Dolgov
et al. (2007 and 2011).  The joint research programme began in 1987, initially collecting stomach samples of cod
and haddock with the objectives of quantitative analysis of demersal fish stomachs, calculating consumption by
cod of commercially important prey species, and creating the basis for developing Barents Sea multispecies
models (Dolgov et al., 2007).  Since the surveys began, other species have been included to further understand
trophic interactions.  The stomach samples are taken on research surveys that use both pelagic and bottom trawls.
Up to ten stomachs are collected for each 10 cm length group at stations which have biological sampling on
Norwegian surveys  (Mehl and Yaragina, 1992).  On Russian commercial vessels and Russian national surveys,
stomachs are sampled per trawl.  Unlike the historical UK surveys in the Barents Sea, these stomachs are
weighed and the total weight and degree of digestion for each prey item is recorded.  For items that can be
identified and intact, lengths are recorded, as well as the total number of identifiable prey in each stomach.
Maturity and sex are also recorded, and otoliths read to measure age.  Only the Norwegian data is included in the
Barents Sea cod dataset, published alongside this paper.

**2.3 Barents Sea cod dataset**
The UK stomach contents dataset has been merged with the Norway data from 1984 as part of the project
CoDINA, to form  the Barents Sea cod dataset.    As part of the merging process, data underwent a thorough
quality control, as described in Holt et al., (2018).   A description of each prey category is provided in
Supplementary material 1, and the metadata for the dataset is provided in Supplementary material 2.  *Data*
*summary*
The largest number and geographic spread of samples are from Norwegian surveys, with fewer samples from UK
surveys (Figure 1).  The data includes the area to the west and north of Svalbard (Spitsbergen).
A total of 400,054 individual stomachs are contained in the Barents Sea cod diet dataset (Table 1). These include
102,197 empty stomachs.  The numbers sampled in each year vary according to the number of surveys in each
year, with no stomach data in some years (Figure 2).  The number of empty stomachs varies each year(Figure 2).
The UK qualitative data in Figure 2 are the 103 pooled records in the UK dataset, where the contents of more than
one stomach are recorded together.  Up to 198 stomachs are combined in each of these records.
The Barents Sea cod diet dataset contains data from across the Barents Sea, from the north of Norway, to
Spitsbergen and eastwards to Russia (Figure 1), however the overall coverage and sampling locations varied each
year.  The UK surveys in the 1930s and 1940s tended to be in the region south and east of Spitsbergen, and around
Bear Island. From the 1980s onwards, the Norwegian survey area was further to the eastern Barents Sea (Figure
3).  There is no data in the dataset for the 1970s, as the UK surveys stopped in the 1960s, and the IMR and PINRO
joint collection of quantitative data did not begin until the 1980s (Dolgov et al., 2007).
Stomachs have been sampled throughout the year (Figure 4) allowing for seasonal changes in the diet to be
analysed. Sampling is widespread in quarters 1, 3 and 4, but does not go as far north in quarters 1 and 2. This is
because there is ice cover preventing the survey vessels from travelling north and east of Spitsbergen during the
winter. It is generally more limited in geographical area during quarter 2 as few regular surveys have been carried
out in that quarter.
*Diet composition*
The dataset shows that cod diets do not remain constant, and occurrence of different prey items changes each
decade (Figure 5) and year (Figure 6). In the 1930s, when there are fewer records, most of the food items are not
identified to species, and there is a large proportion of other food and other fish in the diets (Figure 5). From the
1940s onwards, most of the fish items found in the stomach are identified to species. The data shows a large
amount of herring in the diet in the 1930s, which is not found again in later decades. In the 1940s and 1950s, there
is a high occurrence of euphausiids in the diet, and this decreases to the 2010s. There is a lower occurrence of
capelin in the earlier decades, particularly in the 1930s and 1960s, and this increases again to a high proportion of
the diet from 1990s onwards. Cod cannibalism is apparent in every decade, with the highest proportion of cod in
the diet at >30% in 1930s, reduced to 20% or less thereafter in later years. These figures show how variable the
diet compositions are between years and decades. There is a large proportion of herring in the diets in the 1930s,
which does not occur again, and in more recent decades, capelin, invertebrates and unidentified fish (other fish)
dominate.
Looking at the prey occurrence of the main prey items in each year (Figure 6), there are quite large annual
fluctuations, particularly for capelin, cod, euphausiids and shrimp. Haddock, hyperiids, redfish, polar cod and
herring have fewer annual spikes. Capelin, cod, euphausiids and shrimp have the highest frequency of occurrence
in the earlier part of the time series, to the 1960s. The occurrence is still variable from the 1980s onwards, but to
a lesser degree.
The four main prey species of cod (cannibalism), capelin, euphausiids and shrimp were caught across the whole
geographical area of the surveys (Figure 7). All of these species are caught up to the northern limits of the surveys,
around Spitsbergen and across the Barents Sea.
**2.4 Russian data on cod diet in the Barents Sea**
In addition to the joint Norway-Russia research programme, since 1947, a Russian sampling programme has
collected observations on cod diet in the Barents Sea throughout the year from commercial and research vessels.
During sampling, the degree of stomach fullness was recorded using a five-division scale, ranging from 0: empty
stomach, to 4: stomach expanded and unfolded by food, as well as the presence of different prey items (capelin,
juvenile cod, redfish, herring, shrimp, euphausiids, and other) in the stomach. This qualitative method named
"field feeding analysis" was widely used in Russian investigations of different fish species including cod (see
references in Dolgov et al. 2007 and Yaragina and Dolgov, 2011). From 9 to 45 thousand cod stomachs were
analyzed each year during 1947-1979. As yet, the qualitative Russian stomach samples for years 1947-1983 are
not fully digitized, and so only the digitized data are presented in the supplementary material.
There are 24,457 quantitative and 2,599,421 qualitative Russian stomach samples, and the Russian data extends
further east and northeast than the Norwegian or UK data. The Russian data are not available for publication but
are described and presented in a number of papers and reports (e.g. Zatsepin and Petrova, 1939; Mehl and
Yaragina, 1992; Dolgov et al., 2007 and references therein; Yaragina et al., 2009; Yaragina and Dolgov, 2011
and references therein; Holt et al., 2019). They are available under joint research projects. Further information
about the Russian data is provided in Supplementary material 3. The locations of the samples are shown in
Figure S3.1, the location in each decade in Figure S3.2, the total number of stomachs in each year, and empty,
are in Figure S3.3, the percentage occurrence of prey in each decade in Figure S3.4, and the time series of
occurrence of the main prey is provided in Figure S3.5.
Analysis of the early Russian data also shows that the diets of cod have changed considerably from the 1930s to
the 2000s (Yaragina et al., 2009; Yaragina and Dolgov, 2011), reflecting the trends seen in the Barents Sea diet
database for herring, cod, capelin and polar cod in Figures 5 and 6, although not for haddock. The earliest Russian
investigations into cod diets from the 1930s (Zatsepin and Petrova, 1939) show similar fluctuations in prey, with
interannual asynchronous fluctuations in capelin and euphausiids (Yaragina and Dolgov, 2011), which is also
shown in the data in Figure 6.

**3 Discussion**
IMR/PINRO data have been used in numerous publications and assessments, such as Holt et al. (2019) who
investigated how cod diet changes over time, across seasons and with ontogeny. The role of macroplankton in the
diet has been studied by Orlova et al. (2005). The data were used to extrapolate cod cannibalism information back
to the 1940s (Yaragina et al., 2018). Furthermore, these data were used to explore intra-and inter-specific
interactions between top predators in the Barents Sea (Durant et al. 2014). The Arctic Fisheries Working Group
has used the cod diet data to estimate cod predation on North East Arctic cod and haddock and Barents Sea capelin
in their stock assessments (ICES, 2019). Spatial dynamics of cod and their main prey were determined by
Johannesen et al. (2012), and seasonal variations in feeding and growth by Johannesen et al. (2015). The role of
herring and capelin as prey sources have been studied in detail, particularly in relation to size-dependent predation
(Johansen, 2002; 2003; 2004). The stomach data have also been used to assess Ctenophora abundance in the
Barents Sea, by using cod as a Ctenophora sampling tool (Eriksen et al., 2018). They found that Ctenophora are
increasing abundance in cod stomachs in recent years, coinciding with warm seas. The UK dataset covers the
period of the 1940s when temperatures in the Barents Sea were similar to those found today (Boitsov et al., 2012).
Analysis of this earlier dataset has shown how prey choice is influenced by temperature, with implications for the
present day cod population (Townhill et al., 2016). By combining the early and recent years, this new long-term
dataset will allow further comparison of temperature regimes throughout the past century. Also, by using cod as
a sampling tool, the data can be used to investigate occurrence and trends in any of the species on which they
prey. This has been done e.g. by Holt et al. (2021) for cod predation on snow crab (*Chionoecetes opilio*) which is
a newly established species in the Barents Sea. UK data has been used to investigate diets in the last century and
the role of sea temperature (Townhill et al., 2015). This analysis of the UK data alone found that temperature has
a large role to play in explaining the presence of capelin and herring in cod diets. The Russian data were very
useful for the understanding of the fluctuations in the ecosystem (e.g. Yaragina and Dolgov, 2011) and for the
development of multispecies models. By combining these datasets, we can further understand how the
environment and ecosystems are responding to climatic changes, and what influences the diet and prey switching
of cod which is evident in the data. Such a long time series will enable trends in temperature and variability
indices to be tested against the occurrence of different prey items, and investigate whether fishing pressure on cod
and the stocks of their prey affect the diet composition. The dataset will also enable us to improve parametrisation
of food web models, and to forecast how Barents Sea fisheries may respond in the future, to management and to
climate change.
**3.1 Limitations**
The UK data contains pooled data of up to 198 stomachs in one record, where the stomach data for all of the cod
at one sample station was recorded as one record. This data can be used for qualitative analysis, and exploratory
analysis of the first half of the 20th century. The stomach contents from the pooled data have been previously
presented by Brown and Cheng (1946). The UK data are not as robust as more recent data in that a statistically
designed survey was not carried out, and instead the vessels sought the highest catches of cod that they could.
This must be taken into account in any analysis of the dataset, but nonetheless the data is still valuable and is a
record of cod diets in a certain place and time. There is more detail included in the Norwegian-Russian dataset,
such as fullness of stomachs and length and weight of prey. Where such information is required in analysis, the
UK data may be less useful. However, there is a lot of value in the combined dataset, even with fewer parameters
recorded for the earlier years. The UK data shows similar trends in cod diet to quantitative Russian data for the
same time period (Yaragina and Dolgov, 2011), showing that this qualitative data is still useful in investigating
trends in cod diets.
The quantitative Norwegian and Russian data is more robust than the UK data, and full details of the sampling
methods are available (Dolgov et al., 2007, 2011). The main limitation is that bottom trawls are generally used
and so the cod are not well sampled if they are high in the water column. However, cod are generally a demersal
species and as such bottom trawling is the most effective sampling method. Also, the sampling is limited in the
Lofoten/Vesterålen area, an important spawning location for Barents Sea cod. Analysis of the stomachs of
spawning cod has only been possible for certain years, owing to the low number of survey stations in the area
(Michalsen et al., 2008). As such, cod stomachs sampled south of 70°N and west of 18°E (Lofoten and nearby
areas), were excluded from the dataset and our analyses, as spawning cod is mainly found in this coastal area
(Michalsen et al., 2008). This analysis showed that herring dominated the diet and stomach fullness was found to
be lower in this area during the spawning period (March and April). As such, the location of the cod should be
considered when using this Barents Sea cod diet dataset.

**4 Summary**
The release of the Barents Sea cod diet dataset is a significant contribution to the study of Atlantic cod ecology,
feeding and the Barents Sea ecosystem as a whole. The data have been used in numerous analyses, which has
helped scientists gain a detailed understanding of the stock, mainly analysis of separate datasets. Now, with the
population at a high level, this combined dataset, covering almost 90 years and stretching back to 1930, can be
used to investigate how climate may be affecting the dynamics of the stock, how this may have knock-on effects
within the food web, and what implications this may have for the future of this ecologically and economically
important cod stock.
**Data Availability**
The Barents Sea cod diet database (Townhill et al., 2020) can be accessed and data downloaded from
https://doi.org/10.21335/NMDC-2139169383. The prey categories and metadata for the database are found in

Supplementary material 1 and 2 respectively. The Russian quantitative data from the joint database (1984-2018) and the qualitative Russian diet data (1947-1983), which are not yet fully digitized, are not publicly available due to the Institution policy, but access to these data is granted through contracted collaboration in joint projects with the Polar branch of VNIRO. Summaries, descriptions and analyses of the Russian data can be found in the following publications: Zenkevich and Brotskaya, 1931; Zatsepin and Petrova, 1939; Mehl and Yaragina, 1992; Dolgov et al., 2007; Yaragina and Dolgov, 2011; Holt et al., 2019; Yaragina et al., 2009; and Yaragina and Dolgov, 2011.

**Author Contribution**

BLT conceived the idea for the paper. BB, EJ, NY, AD were all involved in data collection and survey organisation. REH and BLT formed, cleaned and prepared the new Barents Sea Cod Diet Database. REH and BLT prepared the figures for the manuscript. BLT wrote the manuscript with contributions from all co-authors.

**Competing Interests**

The authors declare that they have no conflict of interest.

## 5 Acknowledgements

This research was supported by The Research Council of Norway (RCN), through a MARINFORSK grant "CoDINA—Cod: Diet and food web dyNAmics" (Project No: 255460). GO was also supported by a grant from the European Research Council through the H2020 'Integrated Arctic Observation System' (INTAROS) project (No.727890). Digitization of the UK data was also supported by Cefas Seedcorn project DP332 Trawling Through Time. The authors acknowledge the contribution of all those involved in design of these surveys and data collection, across Norway, Russia and the UK.

We further thank everyone involved in initiating, establishing, and updating the joint Norwegian-Russian and the UK stomach content databases, not least the colleagues undertaking the enormous practical task of identifying the stomach-contents.

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

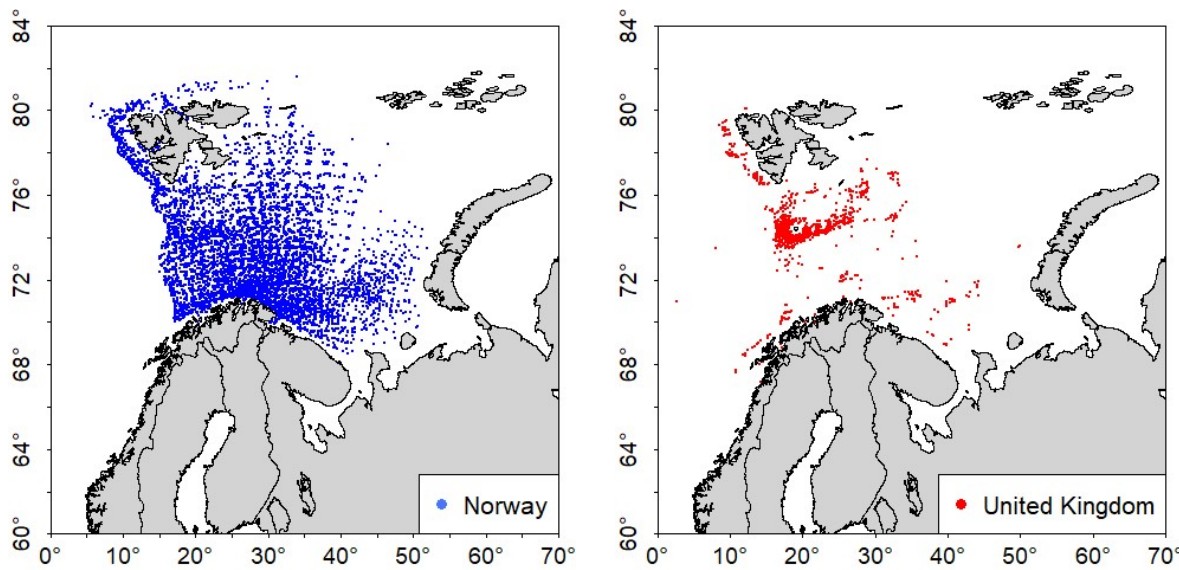

**Figure 1. The location of the cod stomach samples taken in the Barents Sea by each institution.**







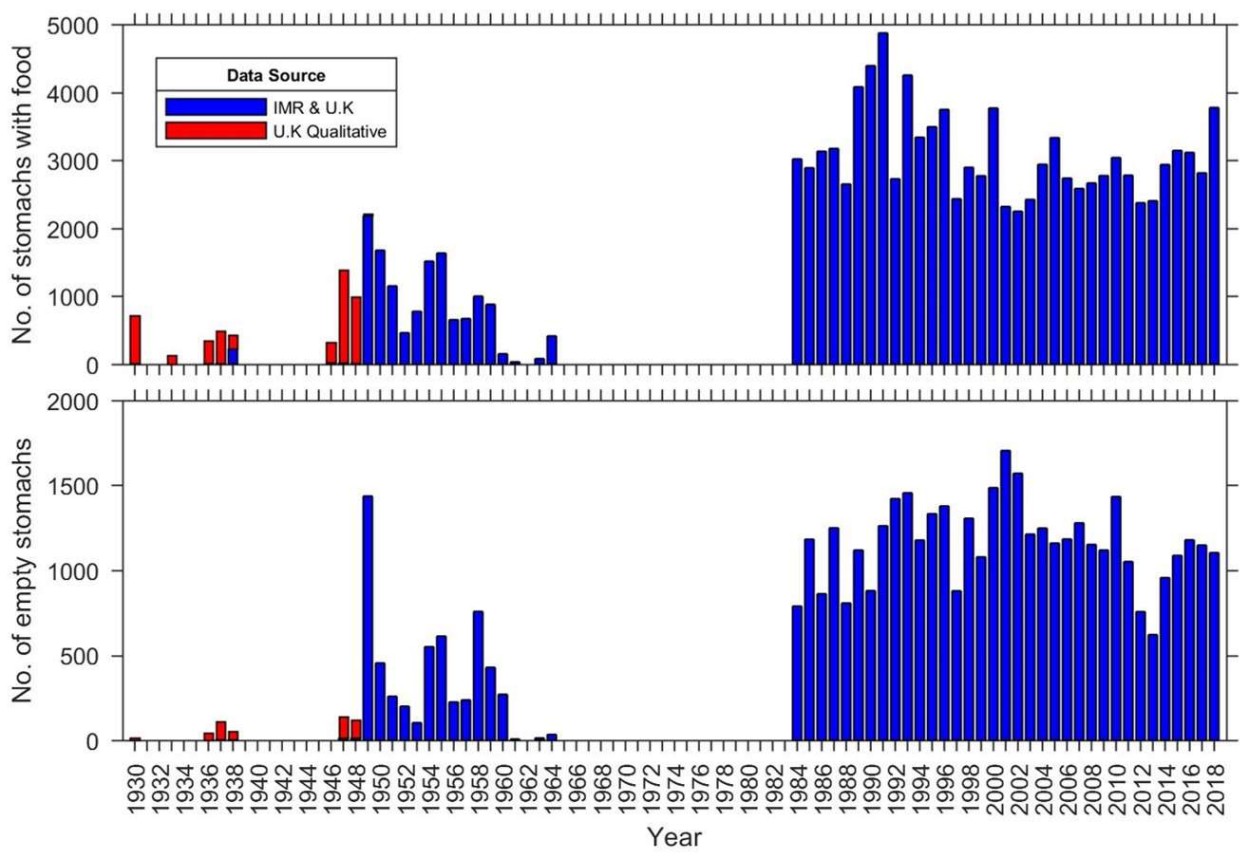


**Figure 2. Number of stomachs sampled in each year, showing those with food contents (upper panel) and those that**
**were empty (lower panel).**







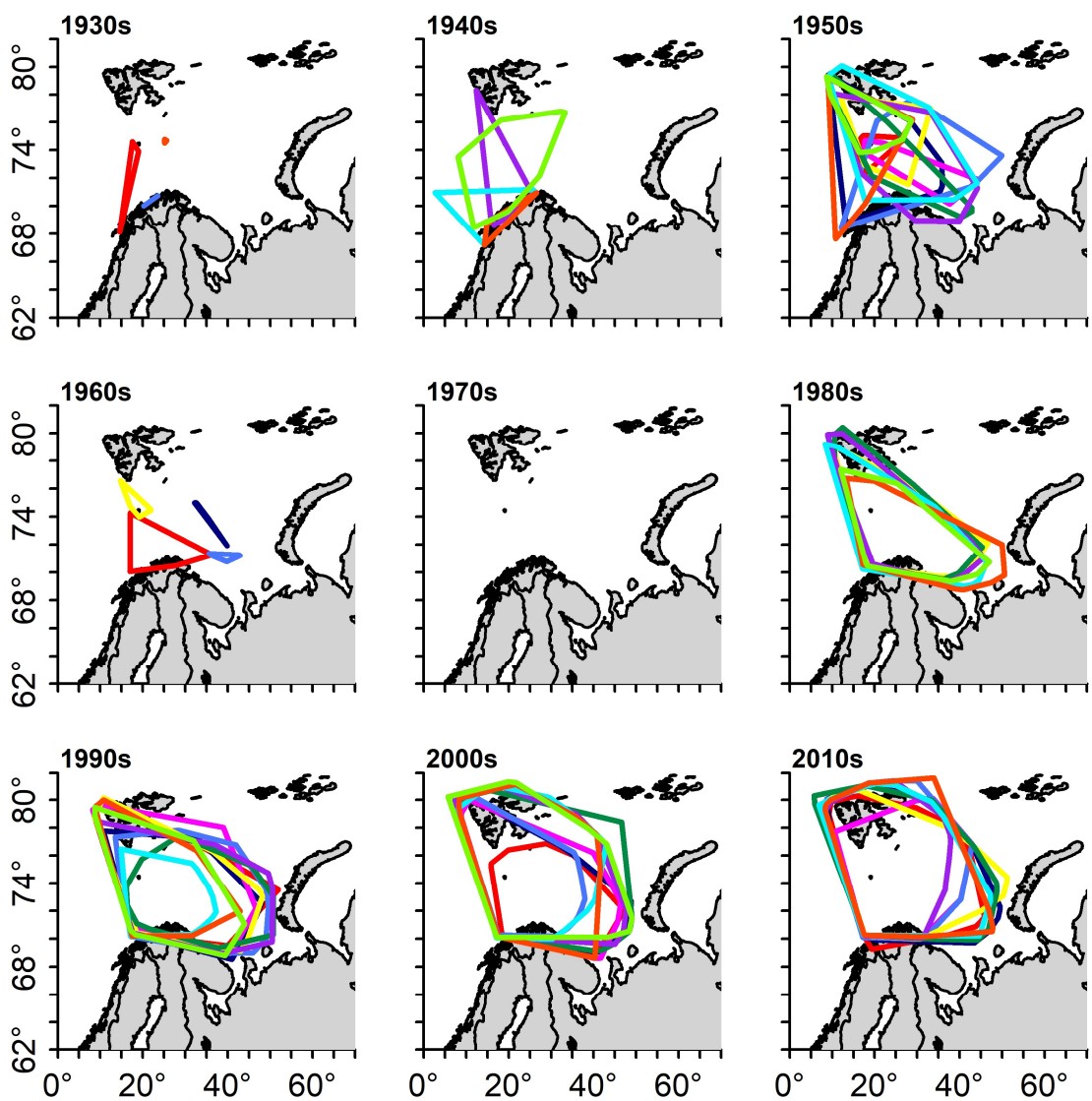

Figure 3. Extent of the sampling coverage in each decade. Red = year 0 e.g. 1940, dark blue = year 1 e.g. 1941, pink = year 2 e.g. 1942, navy blue = year 3 e.g. 1943, yellow = year 4 = 1944, dark green = year 5 e.g. 1945, purple = year 6 e.g. 1946, pale blue = year 7 = 1947, orange = year 8 e.g. 1948, pale green = year 9 e.g. 1949.

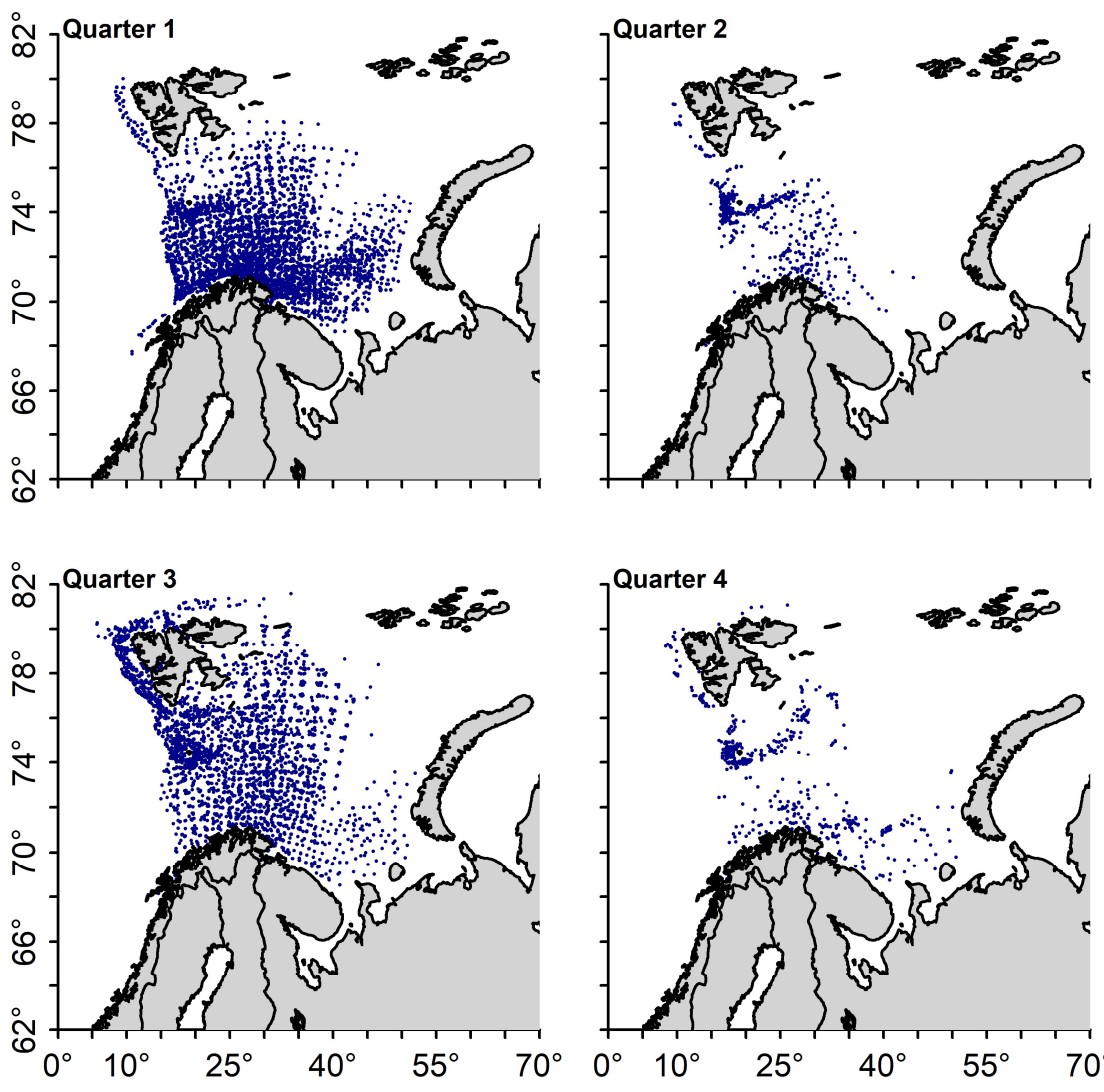

**Figure 4. Sampling coverage in each quarter over all years combined. Each dot denotes a stomach sample.**


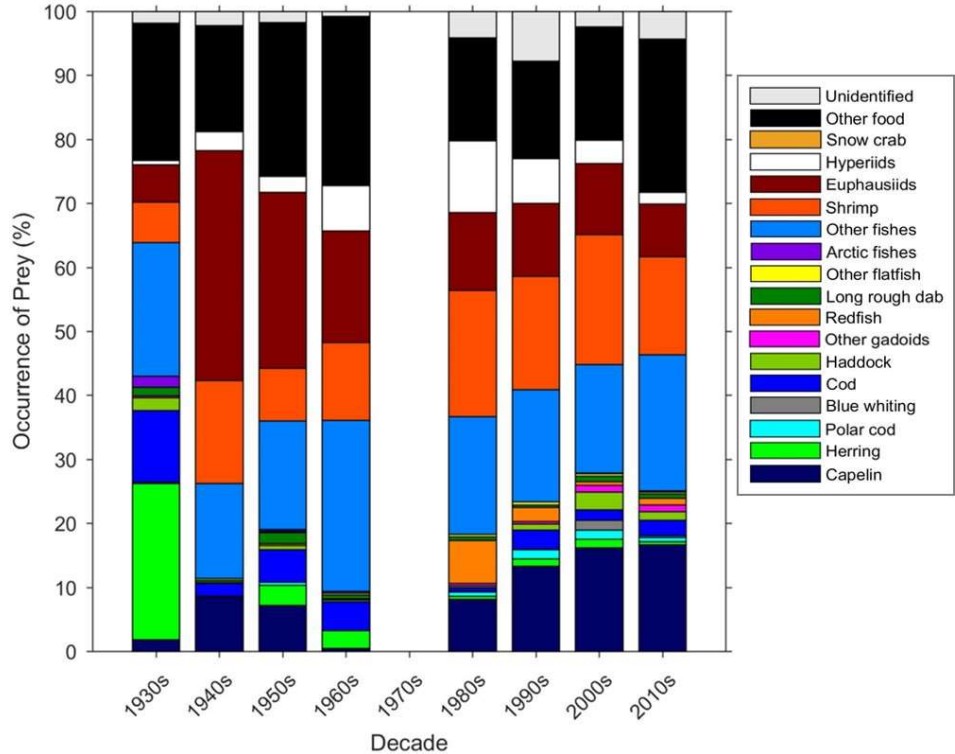


**Figure 5. The percentage occurrence of prey in each decade. The percentage occurrence of each prey item is calculated based on the total prey items in each decade and excludes empty stomachs.**


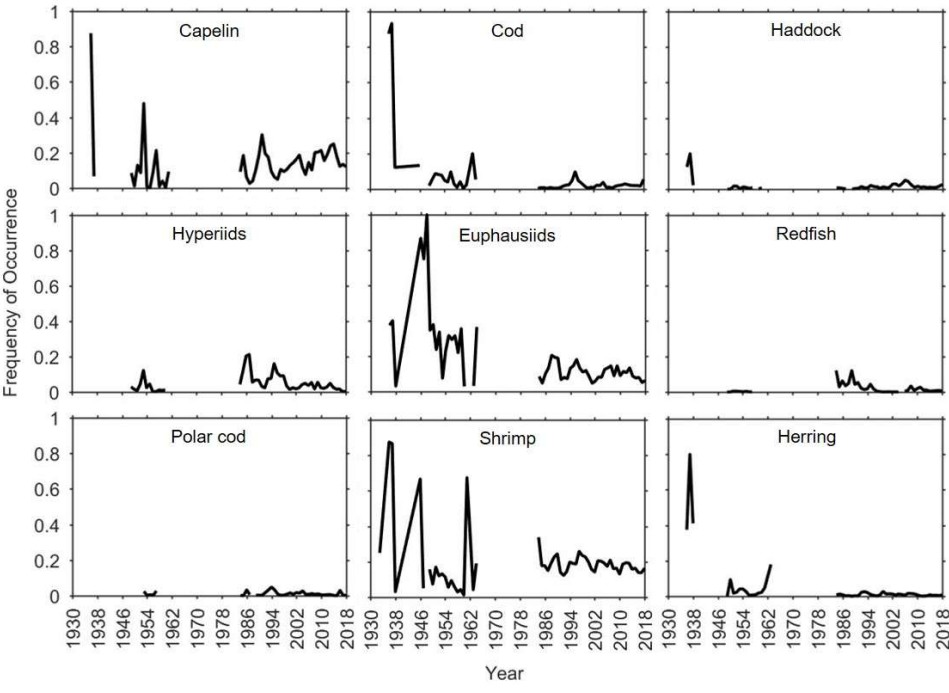


**Figure 6. Time series of occurrence of the main prey items in the dataset, excluding empty stomachs. The frequency of occurrence of each prey item is calculated based on the total number of stomachs in each year.**

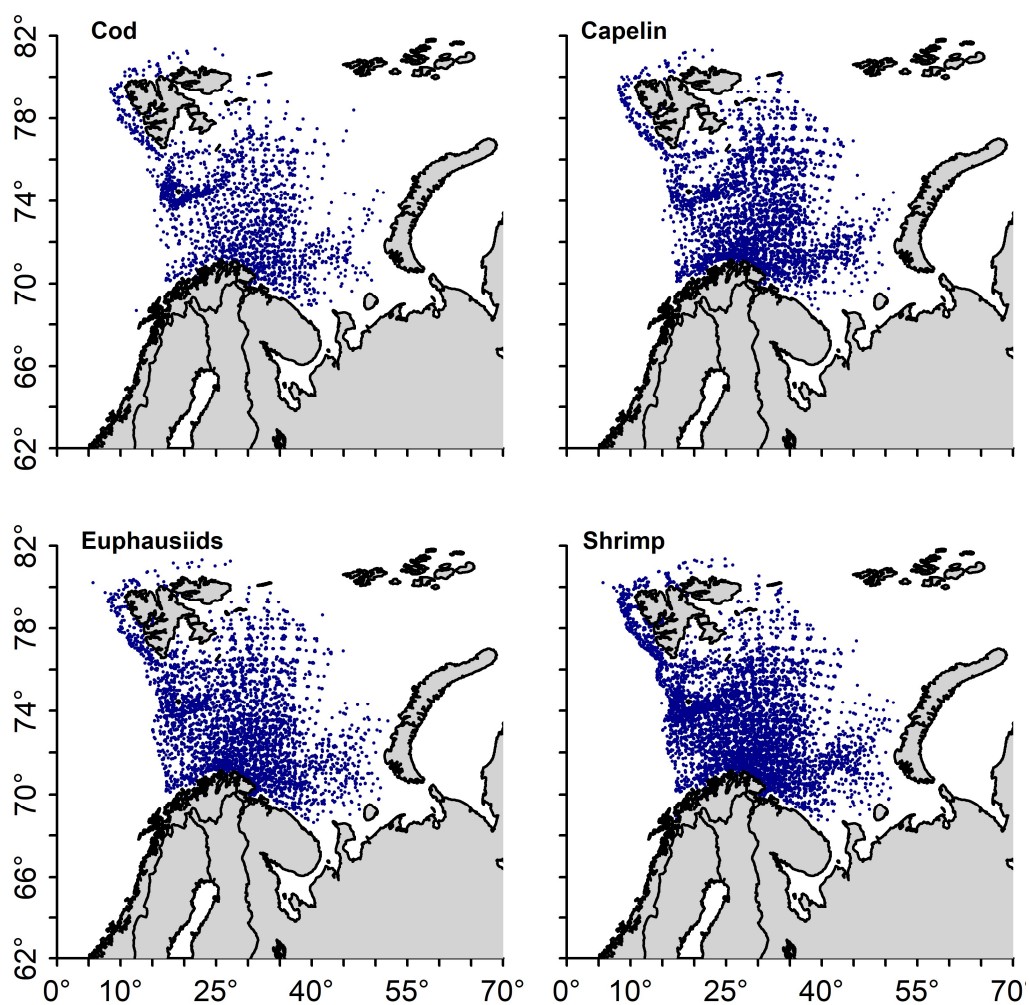


**Figure 7. The presence of the main prey species in stomachs over all years combined. Each dot denotes a stomach**
**sample.**


8 Table
**Table 1. The data available on Barents Sea cod stomachs**

| Source | Years | Total number of stomachs (Incl. Empty) | No. by quarter | Empty stomachs ( number, n) | Area | Pooled or single stomachs | Fishing gear | Main prey species | Published in Barents Sea cod diet dataset |
|---|---|---|---|---|---|---|---|---|---|
| UK | 1930-1949 | 103 records totalling 4532 stomachs | Q1: 263 Q2: 685 Q3: 2235 Q4: 1349 | Unknown | Western Barents Sea, focused on Bear Island and Spitsbergen | Pooled | Commercial trawls | Euphausiids, shrimp, fish | Y |
| UK | 1930-1964 | 19003 | Q1: 2935 Q2: 6314 Q3: 4159 Q4: 5595 | Q1: 850 Q2: 2498 Q3: 656 Q4: 1586 | Bear Island, Spitsbergen | Single | Otter trawl | Euphausiids, shrimp, cod, capelin, herring | Y |
| Norway | 1984-2018 | 146 360 | Q1: 85 644 Q2: 6343 Q3: 49 032 | Q1 26 723 Q2: 2079 Q3: 10 599 | Western and central Barents Sea | Single | Pelagic, bottom and commercial trawl | Cod, capelin, shrimp, euphausiids | Y |

| | | | Q4: 5341 | Q4: 1238 | | | | | |
|---|---|---|---|---|---|---|---|---|---|
| Russia | 1986-2018 | 234 587 | Q1: 26 274<br>Q2: 42 933<br>Q3: 60 638<br>Q4 104 742 | Q1: 5970<br>Q2: 14 162<br>Q3: 8339<br>Q4: 27 453 | Western, eastern and central Barents Sea | Quantitative | Pelagic, bottom and commercial trawl | Shrimp, euphausiids, capelin, other fish, hyperiids | N |
| Russia | 1934 - 2018 | 3 304 134 | Not available | n= 709 112 | Western, eastern and central Barents Sea | Qualitative | Pelagic and bottom trawl | Capelin, euphausiids, shrimp, cod | N |
