# Peer review of "Diets of the Barents Sea cod (Gadus morhua) from the 1930s to"

_Earth System Science Data, 2020_

## Short Comment (SC1) · 17 Jul 2020

The correct name for the fourth author is Joël M. Durant. The ORCHID link is correct.
Joël

---

## Author Comment (AC1) · 17 Jul 2020

Thank you for your comment Joel. This will be corrected in the final version.

---

## Referee Comment (RC1) · Anonymous Referee #1 · 16 Nov 2020

**Ms. Ref. No. Essd-2020-96 "Diets of the Barents Sea cod from the 1930s to the present day" by Bryony L. Townhill, Rebecca E. Holt, Bjarte Bogstad, Joël M. Durant, John K. Pinnegar, Andrey V. Dolgov, Natalia A. Yaragina, Edda Johannesen, Geir Ottersen. https://doi.org/10.5194/essd-2020-96**

*General comments*

The manuscript by Townhill et al. describes a unique time series of Barents Sea cod diet data stretching from 1930 to 2018. This is a great source of information and in this manuscript, the data is presented to the scientific community and the public in general. The data sources are described, the applicability and usefullness and some results are presented and discussed. The data set and this manuscript are of great interest to the public and should be considered for publication. However, the authors should spend a little more time on presenting the data overviews, so that the reader gets more information about the data set without going into the analysis of the data.

- In general, an overview table presenting the data sources, years, number of stomachs etc. would be very helpful. For example:

| Source | Years | Total no. of stomachs | No. by quarter | % empty stomachs (or absolute) | Area (either ICES or „natural" e.g. Svalbard, Bear Island etc. | Comments – pooled data, single stomachs | Etc. |
|---|---|---|---|---|---|---|---|
| UK | 1930-1964 | XY | | | | | |
| … | | | | | | | |
| .. | | | | | | | |

- The sampling coverage is presented by quarter. It would be very nice to have at least figure 5 (eventually also for selected prey species/categories in figure 6) for each decade split up by quarter.
- The authors should avoid stating "The location of each stomach sample is shown in Figure 1", but state what can be seen or deduced from the data and then refer to the figure in parentheses. This applies to the entire document.

*Specific comments*

L1: Title – Reading this title, one question comes to my mind immediately. Will you update this data set regularly? If not, then phrasing "…to the present day" could be a bit misleading, let us say in 10 years of time. So maybe, just indicating the latest year of data (2018) is more appropriate. However, this is just a recommendation. And maybe I am old fashioned, but in my opinion you should have the Latin name of cod in the title.

L39: "conducted" sounds better than "done"

L70: "As part of the merging process, the data underwent a thorough quality control." Either you should refer to a publication specifying this quality control or you have to describe this in the data and methodology section.

L75/76: Please add a few sentences elaborating why understanding trophic interactions in marine ecosystems is important -> e.g. multi-species assessments.

L142: "For items that can be identified, lenghts are recorded, …" – I assume that you refer to "For items that can be identified *and were assessed as being intact (digestion grade 1, eventually 2)*, lengths are recorded,…", because it is possible to identify stomach items, based on fragments, where no length measurement is possible.

L152/153: Does this sentence mean that the missing Russian data for years 1947-1983 will be digitized and added to the data set. Based on the Russian data policy most probably not, but I am just curious if they will become available in the future.

L223: "were" instead of „are"

Figure 2: In the two lower graphs, the factors "x $10^5$" and "x $10^4$" should be placed somewhere else, e.g. "No. of stomachs with food (x $10^5$)"

Supplementary material 2: In the first paragraph, the authors describe the presented table and refer to the column numbers, e.g. "predator information (columns 1-12)". For convenience it would be nice if you could add one column on the left hand side of the "Column name", so that the reader can find the different columns more easily.

| Column No. | Column Name | Information | Units |
|---|---|---|---|
| 1 | Data_ID | Details whether the data is from either the IMR-PINRO joint database or CEFAS | - |
| 2 | Ser_No_Fish | Serial number for each individual fish | - |
| 3 | Country | Country code denoting either:
58: Norway (IMR-PINRO Database)
U.K: CEFAS Data | - |
| 4 | Ship_code | Ship identification code | - |
| … | Year | | |

---

## Editor Comment (EC1) · Dirk Fleischer (Editor) · 3 Dec 2020

Dear authors, according to the now long lasting difficulties to find reviewers, I add this review on your manuscript.

Within your manuscript you describe and present a very impressive dataset spanning the period of 90 years of fishery ecology and feeding behaviour. The data from stomach analysis are tremendously valuable to the ecological community and modeling community. While the diet of cod as a very important species within the Barents Sea system is a quantified connection between the components of this ecosystem and the possible effects any change in stock size might have on the directly connected components or the system as a whole. This dataset it valuable and should be published after some

revision. However the value of the data set should be emphasized with some more detailed informations on the dataset and its usability. Bringing the data to the for ground and having it ready for reuse is right at the hart of ESSD. The availability of the data is great a CC4 license and a flexible landing page at the repository directly pointing to the download of the dataset, well chosen.

General comments: A table breaking the most important key data about the data set into digestible format would be great. Consider time slices as rows per dataset combined with the country of origin. Columns could be years or period, total number of samples, % empty stomachs, area of sampling (ICES rectangles or natural areas), Pooled data or individual samples, and other fields such as most important prey species or average number of prey species, fishing gear, ......

Within the section 2 of the text you present very well the methodology for the UK dataset. You do mention the fishing gear for that particular dataset (actually the oldest set, but you do not present this important information for the other datasets. I also have to say, that the structure of the section 2 is for me while reading a kind of puzzling. May be the already mentioned table and/or the leaving the Russian data out of the description at all could bring some clarity or structure to the text as well. Please find a structure what you think is required to describe the datasets by and then keep to the structure in all subsections describing all the individual datasets.

The Russian data are not part of the published dataset and they are not available at all by the here publishing authors. Why are these data then mentioned to such great extent within the manuscript? Mentioning the data is important and a valuable information to the reader where to find more data if accessible. Just keep the focus more to the published dataset and not dilute the message of the published data by mentioning the unavailable data.

The figures presented to describe the datasets could have more resolution to increase information content. The Russian data could be removed or colored differently in the

figures.

It would be nice to have representation of the most important prey species distribution maps

Figure 2 the scales at the y axis are all in the same order of magnitude, but drawn differently, this should be avoided

Figure 3 could include information per time period on the ice extent, which would give some information on why there are only in the later years data from the east of Svalbard Is there a reason in figure 3 why the maps represent the decades and no other time periods? Anyway, it is good to stick to the chosen periods in all figures, which you did in the current status of the manuscript, but I would consider a different spacing in time or changing from points to like convex hulls to represent the sampling area to make the extent of the data more prominent Since all the points are not distinguishable from each other either figure size or type could help.

Maps on the stomach content could be summarized by pie charts representing the ICES rectangles this would also reveal if there were areas of increased number of empty stomachs.

All these comments and suggestions do cause number of changes to the manuscript and I suggest that i will have a more detail oriented look at the manuscript after these revisions, which I consider only minor revisions.

Regards Dirk Fleischer
* * *

---

## Author Response (AR1)

Ms. Ref. No. Essd-2020-96 "Diets of the Barents Sea cod from the 1930s to the present day" by
Bryony L. Townhill, Rebecca E. Holt, Bjarte Bogstad, Joël M. Durant, John K. Pinnegar, Andrey V.
Dolgov, Natalia A. Yaragina, Edda Johannesen, Geir Ottersen. https://doi.org/10.5194/essd-2020- 96

General comments

The manuscript by Townhill et al. describes a unique time series of Barents Sea cod diet data
stretching from 1930 to 2018. This is a great source of information and in this manuscript, the data is
presented to the scientific community and the public in general. The data sources are described, the
applicability and usefullness and some results are presented and discussed. The data set and this
manuscript are of great interest to the public and should be considered for publication. However,
the authors should spend a little more time on presenting the data overviews, so that the reader
gets more information about the data set without going into the analysis of the data.

Thank you for your comments. We have amended the manuscript as suggested and hope that the
data description is now more detailed, particularly with the addition of the table.

☐ In general, an overview table presenting the data sources, years, number of stomachs etc. would
be very helpful. For example:

| Source | Years | Total no. of stomachs | No. by quarter | % empty stomachs (or absolute) | Area (either ICES or „natural" e.g. Svalbard, Bear Island etc. | Comments – pooled data, single stomachs | Etc. |
|---|---|---|---|---|---|---|---|
| UK | 1930-1964 | XY | | | | | |
| … | | | | | | | |
| .. | | | | | | | |

A table has been included as suggested. Table inserted at end of manuscript and reference to it in
Data summary section.

☐ The sampling coverage is presented by quarter. It would be very nice to have at least figure 5
(eventually also for selected prey species/categories in figure 6) for each decade split up by quarter.

We produced these plots for figures 5 and 6 (below), but they are complicated and do not feel they
add anything to the description of the data.  Making separate plots for each quarter also does not
reveal anything more of the data.  Therefore we propose not to include these plots in the revised
manuscript.  Looking at how the prey change over the years and within each year, and the drivers of
this, is the subject of further analysis on the data, and will be published separately.

[Figure]

[Figure]

The authors should avoid stating "The location of each stomach sample is shown in Figure 1", but
state what can be seen or deduced from the data and then refer to the figure in parentheses. This
applies to the entire document.

This has been amended throughout the document.

Specific comments

L1: Title – Reading this title, one question comes to my mind immediately. Will you update this data set regularly? If not, then phrasing "…to the present day" could be a bit misleading, let us say in 10 years of time. So maybe, just indicating the latest year of data (2018) is more appropriate. However, this is just a recommendation. And maybe I am old fashioned, but in my opinion you should have the Latin name of cod in the title.

The title has now been amended to address both of these.

L39: "conducted" sounds better than "done"

This has been amended.

L70: "As part of the merging process, the data underwent a thorough quality control." Either you should refer to a publication specifying this quality control or you have to describe this in the data and methodology section.

This sentence has now been moved to section 2.4 and a reference included.

L75/76: Please add a few sentences elaborating why understanding trophic interactions in marine ecosystems is important -> e.g. multi-species assessments.

This has now been included in the introduction.

L142: "For items that can be identified, lenghts are recorded, …" – I assume that you refer to "For items that can be identified and were assessed as being intact (digestion grade 1, eventually 2), lengths are recorded,…", because it is possible to identify stomach items, based on fragments, where no length measurement is possible.

This has been amended in text.

L152/153: Does this sentence mean that the missing Russian data for years 1947-1983 will be digitized and added to the data set. Based on the Russian data policy most probably not, but I am just curious if they will become available in the future.

There are further paper records which are not yet digitized, but will be with over the coming years. They won't be added to the dataset that we present, but can be made available on future collaborations. We think it is important to mention that the data exist, as they are an important part of the picture, and are important to include in future analyses. In the Data and methodology section of the text we have made clearer what is and isn't included.

L223: "were" instead of „are"

This has been amended in the text.

Figure 2: In the two lower graphs, the factors "x 105" and "x 104" should be placed somewhere else, e.g. "No. of stomachs with food (x 105)"

After removing the Russian data, the scales are different and so these are no longer needed.

Supplementary material 2: In the first paragraph, the authors describe the presented table and refer to the column numbers, e.g. "predator information (columns 1-12)". For convenience it would be nice if you could add one column on the left hand side of the "Column name", so that the reader can
find the different columns more easily.

This has now been included.

find the different columns more easily.

| Column No. | Column Name | Information | Units |
|---|---|---|---|
| 1 | Data_ID | Details whether the data is from either the IMR-PINRO joint database or CEFAS | - |
| 2 | Ser_No_Fish | Serial number for each individual fish | - |
| 3 | Country | Country code denoting either: 58: Norway (IMR-PINRO Database) U.K: CEFAS Data | - |
| 4 | Ship_code | Ship identification code | - |
| … | Year | | |

Editor comments

Dear authors, according to the now long lasting difficulties to find reviewers, I add this review on
your manuscript.

Within your manuscript you describe and present a very impressive dataset spanning the period of
90 years of fishery ecology and feeding behaviour. The data from stomach analysis are tremendously
valuable to the ecological community and modeling community. While the diet of cod as a very
important species within the Barents Sea system is a quantified connection between the
components of this ecosystem and the possible effects any change in stock size might have on the
directly connected components or the system as a whole. This dataset it valuable and should be
published after some revision. However the value of the data set should be emphasized with some
more detailed informations on the dataset and its usability. Bringing the data to the for ground and
having it ready for reuse is right at the hart of ESSD. The availability of the data is great a CC4 license
and a flexible landing page at the repository directly pointing to the download of the dataset, well
chosen.

Thank you for your comments. We have included your suggestions in the manuscript, with a few
exceptions where we think these are best presented in future papers which analyse the data. We
have described these below.

General comments: A table breaking the most important key data about the data set into digestible
format would be great. Consider time slices as rows per dataset combined with the country of origin.
Columns could be years or period, total number of samples, % empty stomachs, area of sampling
(ICES rectangles or natural areas), Pooled data or individual samples, and other fields such as most
important prey species or average number of prey species, fishing gear, ......

This table has been included as per the previous reviewer's comment.

Within the section 2 of the text you present very well the methodology for the UK dataset. You do
mention the fishing gear for that particular dataset (actually the oldest set, but you do not present
this important information for the other datasets. I also have to say, that the structure of the section
2 is for me while reading a kind of puzzling. May be the already mentioned table and/or the leaving
the Russian data out of the description at all could bring some clarity or structure to the text as well.

Please find a structure what you think is required to describe the datasets by and then keep to the
structure in all subsections describing all the individual datasets.

Further description of the Norwegian methodology has been included in the Data and methodology
section. It wasn't originally included because it is published in other easily accessible papers,
whereas the UK description isn't. However, more information has now been added.

We have now included the table as suggested, and also rearranged the text in this section to make it
clearer what is included and what is not.

The Russian data are not part of the published dataset and they are not available at all by the here
publishing authors. Why are these data then mentioned to such great extent within the manuscript?
Mentioning the data is important and a valuable information to the reader where to find more data
if accessible. Just keep the focus more to the published dataset and not dilute the message of the
published data by mentioning the unavailable data.

The text describing the Russian surveys has been moved to below the paragraph describing the
Barents Sea dataset, to make it clear that these are not included. We do however think that it is
important to highlight that these additional data exist, because they add information on a larger
geographical area, and also the earlier time period. When further analyses are conducted on these
data, it is important that scientists appreciate that there is other information available, that it can be
available to collaborators, and that a lot of information is included in existing publications.

The figures presented to describe the datasets could have more resolution to increase information
content. The Russian data could be removed or colored differently in the figures.

The data from the Russian surveys has been removed from the figures in the main paper, and have
been included in a new supplement. Removing this data has made the figures clearer, as there is less
data, and the resolution has been increased, particularly for the maps.

It would be nice to have representation of the most important prey species distribution maps

Maps have now been included for the four main prey species: cod, capelin, euphausids, shrimp.

Figure 2 the scales at the y axis are all in the same order of magnitude, but drawn differently, this
should be avoided

This is no longer a problem with the new figures.

Figure 3 could include information per time period on the ice extent, which would give some
information on why there are only in the later years data from the east of Svalbard.

We have searched for data on ice extent but feel it would take a significant amount of work to
calculate the average or minimum ice extent for each decade, particularly for the earlier years for
which there is less data available. As such as have not added it to the maps here, but we have
included mention that the surveys were able to go further north during the summer, when the ice
cover was reduced. Further analyses could look at the role that the changing ice might have on
predator and prey interactions, in which case calculating the change in ice in relation to the dates
that these samples would be carried out.

Is there a reason in figure 3 why the maps represent the decades and no other time periods?
Anyway, it is good to stick to the chosen periods in all figures, which you did in the current status of
the manuscript, but I would consider a different spacing in time or changing from points to like convex hulls to represent the sampling area to make the extent of the data more prominent Since all
the points are not distinguishable from each other either figure size or type could help.

Having removed the Russian data from the maps, reduced the dot size and increased the resolution,
these plots are now much clearer and it is easier to distinguish the different sampling locations.  We
produced maps with convex hulls, but don't think that these improve the maps.  Rather, they are
slightly misleading and they make the sampling area larger than it is because of a small number of
sample sites which extend far beyond most of the other sites.  As such, we propose to include the
new maps with a higher resolution but without the convex hulls.

[Figure]

Maps on the stomach content could be summarized by pie charts representing the ICES rectangles
this would also reveal if there were areas of increased number of empty stomachs.

We have produced the below map of the distribution of empty stomachs per decade.  You can see
that the location of the empty stomachs closely resembles the location of the sample stations, i.e.
the empty stomachs are dispersed amongst the majority of the samples.  As such, we do not think
that this figure adds value to the paper.  The maps that have been included on the extent of the four
main prey species, also show that prey are widely dispersed across the area, rather than each being
concentrated.  However further work can look at the reasons for the empty stomachs or certain
prey, such as time of year (there are seasonal cycles of feeding), whether they coincide with years of
low biomass in prey species etc., productivity, links with NAO etc.

[Figure]

All these comments and suggestions do cause number of changes to the manuscript and I suggest
that i will have a more detail oriented look at the manuscript after these revisions, which I consider
only minor revisions.

Regards Dirk Fleischer

[revised manuscript text omitted]

---

## Author Response (AR2)

Ms. Ref. No. Essd-2020-96 "Diets of the Barents Sea cod from the 1930s to the present day" by
Bryony L. Townhill, Rebecca E. Holt, Bjarte Bogstad, Joël M. Durant, John K. Pinnegar, Andrey V.
Dolgov, Natalia A. Yaragina, Edda Johannesen, Geir Ottersen. https://doi.org/10.5194/essd-2020- 96

Dear Authors,
sorry for the delay in the process. Thank you for your patience.
You improved your manuscript accordingly to the reviews and the minor changes are just a follow up on your own suggestions in your answer of the review comments. As suggested you added convex hulls to the maps of stomach distributions and I do follow your suggestion, that in this form the convex hulls do not provide any additional value to the maps.
The suggestion in the comment may be misleading or poorly described, but for each of the maps for the 10 year period it might be valuable to add convex hulls for each year instead of a huge cloud of individual points. This visualization would include the time as a dimension in the maps and it would also make perfect use of the separated periods of time.

Please consider this last comment for improvement.

Dear editor,

Thank you for your clarification regarding the convex hulls. We have edited the figure so that it now has the convex hulls rather than the dots for each station. We still have reservations that this is the best way to present the data, because the convex hulls do not give an idea of the density of sample stations, and outlying stations can make the area look misleadingly large. However, we are happy for you to include this version of the figure in the paper if you think it shows the annual changes better.

Regards,

Bryony and co-authors.